# COVID-19 Breakthrough Infections in Vaccinated Kidney Transplant Recipients

**DOI:** 10.3390/vaccines10111911

**Published:** 2022-11-11

**Authors:** Xiaojing Zhang, Ruopeng Weng, Fei Liu, Yi Xie, Yanyan Jin, Qiuyu Li, Guoping Huang, Junyi Chen, Jingjing Wang, Huijun Shen, Haidong Fu, Jianhua Mao

**Affiliations:** 1Department of Nephrology, The Children’s Hospital, Zhejiang University School of Medicine, National Clinical Research Center for Child Health, Hangzhou 310052, China; 2Department of Gynecology and Obstetrics, Women’s Hospital, School of Medicine, Zhejiang University, Hangzhou 310007, China

**Keywords:** breakthrough infection, COVID-19, SARS-CoV-2, kidney transplant recipients, vaccine, booster doses, immunogenicity

## Abstract

Coronavirus disease 2019 (COVID-19) is associated with increased morbidity and mortality among kidney transplant recipients (KTRs). The administration of the severe acute respiratory syndrome coronavirus 2 (SARS-CoV-2) vaccination is the only reliable strategy to prevent COVID-19 and alleviate the severity of COVID-19 in this particular population. The aim of this article was to evaluate the clinical protection by vaccines (breakthrough infections, deaths, and hospitalizations) in KTRs. There were 135 KTRs with COVID-19 breakthrough infections for whom patient-level data were available in PubMed and Web of Science. There was a male predominance (61.4%), 97 were given the standard vaccination regimen, and 38 received three or four doses of the vaccine. The median age was 59.0 (IQR: 49.0–69.0) years. A total of 67 patients were hospitalized, and 10 patients died. In 72.6% of cases, triple-maintenance immunosuppression was employed. The deceased patients were older than the survivors (*p* < 0.05); an age over 60 years was a risk factor for death (*p* < 0.05). The KTRs with booster vaccines had a longer time interval from the last vaccine to COVID-19 infection and lower hospitalization rates than the individuals who received the standard vaccination regimen (33.3% vs. 54.8%, *p* < 0.05). The hospitalized patients were older than the outpatients (*p* < 0.05). Among 16,820 fully vaccinated or boosted KTRs from 14 centers, there were 633 breakthrough infections (3.58%) and 73 associated deaths (0.41%). The center-level breakthrough infection rates varied from 0.21% to 9.29%. These findings highlight the need for booster doses for KTRs. However, more research is needed to define the long-term effectiveness and immunogenicity of booster doses and to identify methods to boost the protective response to vaccination in these immunocompromised patients.

## 1. Introduction

Due to their immunocompromised status, kidney transplant recipients (KTRs) are more prone to a severe and prolonged infection of coronavirus disease 2019 (COVID-19). A mortality rate ranging from 20% to 32% has been reported in KTRs compared with 1% to 5% in the general population [1,2,3]. KTRs have a more rapid clinical progression than the general population. Among hospitalized KTRs with COVID-19, the mortality rate was even higher, at 41%. Among those KTRs who survived COVID-19, 19% sustained renal graft dysfunction and 4% suffered graft losses [2].

Vaccination against severe acute respiratory syndrome coronavirus 2 (SARS-CoV-2) is the only reliable strategy to prevent COVID-19 and alleviate the severity of COVID-19 in particular populations. Vaccination among KTRs was associated with a reduced risk of breakthrough infection when compared to unvaccinated KTRs. A study from the Scottish renal registry showed that SARS-CoV-2 infections were found in 8% of fully vaccinated KTRs but in 14% of nonvaccinated KTRs [4]. Nevertheless, mortality among KTRs with breakthrough infection is also much higher (13.18%) than that among the general population [4]. Over 50% of fully vaccinated KTRs may have required hospitalization with severe SARS-CoV-2 infection in the period before the Omicron variant [5,6].

A low antibody response and cellular response to vaccines are thought to be strongly associated with breakthrough infection. Compared with the general population, KTRs experience lower rates of seroconversion following COVID-19 vaccination. In general, the anti–SARS-CoV-2 postvaccine antibody response against the spike protein has been observed in 4.1% to 62.6% of KTRs following two doses of mRNA vaccines, and a cellular response has been observed in 30% to 64.5% of KTRs following two doses of mRNA vaccines [7,8,9,10,11,12]. Given that two doses of the SARS-CoV-2 mRNA vaccine cause an insufficient immune response rate in KTRs, the vaccination strategy was modified in the USA and Europe in summer 2021. Some trials have reported encouraging results. The immunological response to a third dose of the two main mRNA-based vaccines led to increases in IgG seroconversion from 37.5–40% to 61.2–71% at 4 weeks and T-cell response rates from 51.2% to 70.0% [13,14,15]. Even in KTRs with no seroconversion after two doses of the mRNA vaccine, a third dose induced a SARS-CoV-2 antibody response in 39–49% of the population [16,17].

However, it is still unknown whether these results translate to a higher protective effect of SARS-CoV-2 vaccines. The evidence regarding the real-world clinical effectiveness of COVID-19 vaccination in KTRs is limited. The effectiveness of the clinical vaccine matters to patients and decision makers. It is critical to have a broader understanding of the breakthrough infection rate and influencing factors after vaccination among KTRs.

## 2. Methods

We identified relevant publications through electronic searches in PubMed and Web of Science between 1 Jan 2020 and 31 July 2022. The main keywords searched included “kidney transplantation recipients”, “SARS-CoV-2 vaccines”, “breakthrough infection”, and “COVID-19”. These publications provided complete patient-level data of vaccinated KTRs with breakthrough infections or provided numbers of KTRs who received standard or booster vaccinations (less than 100 cases were excluded), developed breakthrough infections after vaccination, and died. Breakthrough infection was defined as positive SARS-CoV-2 PCR result more than 14 days after full-dose vaccination. Standard vaccination was defined as completion of the recommended dosing regimen of any vaccine (2 doses for mRNA vaccines, 1 dose for Janssen vaccine, or 2 doses for other vaccines) and partial vaccination as having received only 1 dose of an mRNA or other vaccine.

Data are presented as the median (interquartile range [IQR]) for nonparametric data. Categorical variables are listed as counts or percentages. Differences in qualitative variables between groups were analyzed with the χ^2^ test or Fisher’s exact test when one or more expected values were less than five or the data were very unequally distributed among the cells of the table. Continuous variables were analyzed using the *t* test or the Mann–Whitney *U* test, as appropriate. Statistical analysis was performed using IBM SPSS Statistics 26.0 (SPSS, Inc., Chicago, IL, USA) software for MacOS. All tests were two-tailed, and the significance level was defined as a *p* value < 0.05.

## 3. Results

A total of 961 studies were screened, in which 128 underwent full-text review, and 32 studies were included [4,5,6,12,18,19,20,21,22,23,24,25,26,27,28,29,30,31,32,33,34,35,36,37,38,39,40,41,42,43,44,45] (Appendix A). Nineteen were letters, nine were articles, two were case reports, and two were brief communications. The studies were performed in Belgium (one study), Brazil (one study), Canada (one study), Croatia (two studies), Czechia (one study), Denmark (one study), France (one study), Greece (one study), India (one study), Italy (two studies), Israel (one study), Scotland (one study), Saudi Arabia (one study), Singapore (one study), Spain (three studies), and the USA (thirteen studies).

### 3.1. Center-Level of COVID-19 Breakthrough Infection in Vaccinated KTRs

We identified kidney transplant centers that were able to provide information about the numbers of KTRs who received standard vaccine regimens (fewer than 100 cases were excluded), developed breakthrough infections after vaccination, and died from PubMed and Web of Science [4,12,18,19,20,21,22,23,24,25,26,27,28,29]. Among a total of 16,820 vaccinated KTRs from 14 centers (Table 1), 633 developed breakthrough infections (3.94%), and there were 75 associated deaths (0.45%). The mortality rate among the KTRs with breakthrough infections was 11.85% (75 of 633). The center-level breakthrough infection rates varied from 0.21% to 9.29%. In the Denmark center [29], 90.4% of the KTRs received three doses of BNT162b2, and the rest of the KTRs received standard vaccination. Regarding vaccine types, mRNA vaccines are mainly applied in 12 centers, and inactivated vaccines and vector vaccines are mainly applied in the other two centers.

### 3.2. Characteristics of KTRs with Patient-Level Data and Comparison According to Vaccination Status

There have been several case reports and case series involving fully vaccinated KTRs with breakthrough infections in PubMed and Web of Science [5,6,19,20,21,22,25,30,31,32,33,34,35,36,37,38,39,40,41,42,43,44,45]. A total of 135 KTRs with patient-level data were analyzed (Table 2). All patients included were older than 18 years, and the median age was 59.0 (IQR: 49.0–69.0) years. None of the patients had a previous history of symptomatic COVID-19. There was a male predominance (61.4%). The time elapsed from kidney transplantation (KT) ranged from 0.1 years to 32.9 years. A total of 97.8% (132/135) of vaccinated subjects received the mRNA-based vaccine (88 received the BNT162b2 vaccine, 42 received the mRNA-1273 vaccine, and 2 received both). Most patients (*n* = 117, 86.7%) received calcineurin inhibitors (CNIs) (tacrolimus or cyclosporin), 85.0% (*n* = 116) received corticosteroids, 83.0% (*n* = 112) received mycophenolate mofetil/mycophenolic acid (MMF/MPA), and some received belatacept, sirolimus, or everolimus. Triple maintenance immunosuppression was documented in 72.6% (*n* = 98) of cases. The time interval from the last vaccine dose to COVID-19 ranged from 14 days to 351 days. Sixty-seven patients (50.0%) were hospitalized, and ten patients (7.5%) died. 

Of these cases, 97 were given the standard vaccination regimen, while 38 received three or four doses of the vaccine (Figure 1). A comparison between these two groups is shown in Table 3. Among the 38 KTRs with booster vaccines, the median age was 60.5 (IQR: 51.0–72.0) years, the median time from transplantation was 3.5 (IQR: 1.5–7.7) years, and the median time from the last vaccine dose to SARS-CoV-2 infection was 218.0 (IQR: 149.3–267.0) days. Twenty-three patients were male. Thirty-four patients (89.5%) received triple maintenance immunosuppression. Compared with the patients who received the standard vaccination (*n* = 97), the KTRs who received the booster vaccination did not differ in age, sex, time from transplantation, or breakthrough infection-related deaths but had a higher ratio of MMF/MPA (82.1% vs. 76.6%, respectively, *p* < 0.05) and triple immunosuppressive regimens (89.5% vs. 66.0%, respectively, *p* < 0.05), a longer time interval from the last vaccine to COVID-19 (median: 218.0 days vs. 53.0 days, respectively, *p* < 0.05), and a lower hospitalization rate (34.2% vs. 56.3%, respectively, *p* < 0.05).

### 3.3. Comparison between COVID-19 Breakthrough Infection-Related Deaths and Survival

Among the 10 patients who had breakthrough infection-related deaths (Table 4), the median age was 69.5 (IQR: 64.3–71.3) years, and 9 patients were over 60 years of age. The median time from KT was 7.2 (IQR: 2.8–13.8) years, and the median time from the last vaccine to SARS-CoV-2 infection was 69.0 (IQR: 24.0–152.5) days. Nine patients received the mRNA-based vaccine, and one received the ChAdOx1 vaccine. Nine patients received triple immunosuppressive therapy, including MMF/MPA and CNIs. Compared with the survivors, the deceased patients were older (median: 69.5 years vs. 57.5 years, respectively, *p* < 0.05), with a higher proportion of patients over 60 years (90.0% vs. 45.6%, *p* < 0.05). There were no significant differences between the two groups in terms of sex, maintenance immunosuppression regimen (triple maintenance immunosuppression, TAC, or MMF/MPA), the time since the KT, and the time from the last vaccine to SARS-CoV-2 infection.

### 3.4. Comparison between Inpatients and Outpatients with COVID-19 Breakthrough Infections

The characteristics of the inpatients and outpatients with breakthrough COVID-19 infections are detailed in Table 5. Among the 67 hospitalized KTRs, the median age was 65.0 (IQR: 53.0–71.0) years, the median time since transplantation was 3.8 (IQR: 1.6–9.0) years, and the median time from the last vaccine dose to SARS-CoV-2 infection was 74.5 (IQR: 35.0–150.5) days. Forty-eight patients (71.6%) received triple maintenance immunosuppression. Compared with the outpatient cases, the hospitalized patients were older (median: 65 years vs. 53 years, respectively, *p* < 0.05) with a higher proportion of patients over 60 years (65.7% vs. 32.8%, *p* < 0.05) and had a lower ratio of booster vaccination (19.4% vs. 37.3%, *p* < 0.05). There were no significant differences between the two groups in terms of sex, maintenance immunosuppression regimen, the time since KT, or the time from the last vaccine to SARS-CoV-2 infection.

## 4. Discussion

In this study, we analyzed the clinical characteristics of breakthrough COVID-19 infections in vaccinated KTRs. Our results show that KTRs have great vulnerability to COVID-19 even with a booster vaccination. The COVID-19 breakthrough infection rates among vaccinated KTRs are much higher than in the general vaccinated population, ranging from 0.21% to 9.29% (Table 1) versus 0.01% in the general population [46].Of note, the breakthrough infection rate of 0.01% among the general population was reported before the Omicron wave. However, the above higher breakthrough infection rates show that KTRs receive less protection from SARS-CoV-2 vaccination. This may be related to the low antibody response and cellular response to vaccines in KTRs. In addition, KTRs might be more susceptible to infection by SARS-CoV-2 variants of concern (VOCs), especially highly contagious variants such as Delta and Omicron. The extensive immune evasiveness of Omicron resulted in higher breakthrough rates among both the general and immunocompromised populations. As shown in Table 1, the highest central-level breakthrough infection rate of 9.29% was reported during the Omicron wave, even though 90.4% of the patients in this center received a third dose. Excluding the effect of the vaccine type, the breakthrough infection rate was multiplied in the Danish center compared to other centers using mRNA vaccines. This finding suggested that Omicron reduced the efficacy of the current vaccines even with booster doses that have induced higher levels of protection against pre-Omicron variants. 

Anti-spike IgG titers above 143 binding antibody units per milliliter (BAU/mL) correlate with the presence of neutralizing antibodies against the wild-type virus and VOCs [47], and patients with low titers of anti-spike IgG may remain insufficiently protected. Kumar’s research showed that neutralizing antibody positivity in transplant recipients after two doses of the mRNA-1273 vaccine was low against the Alpha, Beta, and Delta variants but subsequently increased with the administration of a third vaccine dose [48]. However, the proportion of KTRs without a sufficient neutralization response against VOCs after a third vaccine dose is still high (41% to 57%), even in patients with seropositivity [49]. A fourth dose of mRNA vaccine was given to 92 KTRs who had anti-spike IgG titers less than 143 BAU/mL 1 month after a third dose. The median anti-spike IgG levels increased from 16.4 BAU/mL to 145 BAU/mL, and 50% of patients reached the threshold of 143 BAU/mL [50]. These studies showed that increasing the number of vaccine doses may improve immune responses in KTRs. Therefore, it is necessary to offer four or more doses to KTRs who have a weak response to previous vaccination.

However, does additional vaccination increase the actual clinical protection rate? Our statistics showed that, compared with KTRs who received a standard vaccination regimen, those who received booster vaccines had a longer time interval from the last vaccine to SARS-CoV-2 infection (median: 218.0 days vs. 53.0 days, respectively, *p* < 0.05) and a lower hospitalization rate (34.2% vs. 56.3%, respectively, *p* < 0.05). This finding supports the notion that booster vaccines confer clinical protection. Consistent with the above results, Benning et al. found that SARS-CoV-2 infection occurred in 12/49 (25%) KTRs at a median of 5.2 months after receiving a third vaccine dose. Notably, all the patients were oligosymptomatic, with no patient requiring hospitalization due to COVID-19 [49]. Cassaniti et al. reported SARS-CoV-2 infection in 11/45 (24.4%) patients at a median of 3.5 months after a third dose. No patient required invasive ventilation or admission to the intensive care unit (ICU), and no deaths were observed [51]. Of note, most (22/23) infections in the two studies occurred in 2022, in parallel with the surge of the Omicron variant in Europe. These observations support the notion of the efficacy of a booster vaccine in protecting against infection and severe disease caused by the variants. Despite impaired protection against an infection, protection against severe disease is maintained. This might be explained by the preservation of T-cell responses against the variants [52]. However, the data on breakthrough infections after three or more vaccine doses in KTRs are limited. The clinical impact of three or more vaccine doses in KTRs with respect to symptomatic COVID-19, hospitalizations, and related deaths is not sufficiently described and should be further investigated, especially in the setting of the current surges related to additional variants as they occur. 

Our results show that up to 56.3% of fully vaccinated KTRs required hospitalization. The mortality rate in the booster group showed no substantial differences compared with the KTRs who received standard vaccination. A recipient age (≥60 years) is one of the main risk factors for COVID-19-associated hospitalization and death. In addition, variable immune responses to additional vaccine doses were observed in this high-risk population. All these findings indicate that other COVID-19-preventive strategies need to be investigated. A short post-kidney transplantation time, poor renal function, triple maintenance immunosuppression, and a regimen that includes MMF/MPA or belatacept were related to a poorer response [7,8,9,10,11,12]. In Reischig’s study, a multivariate analysis showed that short post-transplant periods were associated with a COVID-19 after vaccination [53]. To maximize the likelihood of developing a protective immune response, it is common to wait at least 3 months and usually up to 12 months after transplantation before being vaccinated. In contrast with KTRs, hemodialysis patients more frequently develop specific humoral and cellular responses to SARS-CoV-2 vaccines, almost reaching healthy control levels [9,54]. Therefore, receiving complete COVID-19 vaccination prior to transplantation should be considered for waitlisted patients awaiting transplantation. Treatment with MMF or belatacept was a major determinant of seroconversion failure in KTRs [9,11,55,56,57,58]. KTRs treated with an MMF-free immunosuppressive regimen were 13 times more likely to develop antibodies against SARS-CoV-2 than KTRs treated with MMF [55]. KTRs aged < 60 years without the use of antimetabolites at the time of vaccination showed a very high seroconversion rate (92.9% of cases) [56], very similar to that in the general population receiving the mRNA SARS-CoV-2 vaccine. KTRs treated with belatacept showed a much lower seroconversion rate ranging from 0 to 5.7% after two vaccine doses and 6.4% after a third dose, with low antibody titers [9,57,58], likely to be sustained non-responders even after five vaccine doses [59]. Therefore, the temporary discontinuation or a decrease in the dose of MPA or belatacept may help increase the vaccination response rate. In Osmanodja’s study, pausing MPA and adding a 5 mg prednisolone equivalent for KTRs before the fourth vaccine dose increased the serological response rate to 75% in comparison to no dose adjustment (52%) or a dose reduction (46%) [59]. However, Florina et al. found that in KTRs without a humoral immune response to at least three vaccine doses, pausing MPA/AZA for 2 weeks before an additional vaccination did not increase the rate of seroconversion [60]. In addition, pausing immunosuppressant use may put patients with a high immunologic risk or who have undergone a recent transplantation at a higher risk of allograft rejection [59]. In addition, additional strategies, such as a heterologous vaccination schedule [61] and monoclonal antibody pre-exposure prophylaxis, have been developed regarding KTRs [15,62].

Regarding the safety of the booster vaccines, to date, there are no reported serious adverse events or episodes from the third or fourth dose. However, the safety of a third or fourth dose in KTRs is not clear. Whether a third or fourth dose of vaccine induces the production of new HLA antibodies or enhances the level of existing HLA antibodies in KTRs needs to be studied. In Cassaniti’s study, de novo HLA antibodies appeared in 4 out of 45 (8.8%) KTRs after a third dose, but none of the patients developed acute renal rejection [51].

Our study had some limitations. First, a small number of vaccinated KTRs from few geographical settings were evaluated, which might not be representative of other settings with different epidemiological conditions, in which the efficacy of vaccines might differ. Second, an insufficient number of cases met our inclusion criteria to allow for meaningful comparisons between different vaccine platforms. Third, most of the included studies were published before the emergence and spread of the omicron variant.

## 5. Conclusions

SARS-CoV-2 vaccination is critically important and should be prioritized in all KTRs. A low immune response in KTRs after vaccination is thought to be strongly associated with breakthrough infection. Booster vaccine doses have been proven to significantly increase the rate of seroconversion and clinical protection among KTRs. However, future prospective studies are needed to define the long-term effectiveness and immunogenicity of booster doses and to identify methods to boost the protective response to vaccination in these immunocompromised patients. KTRs face a higher risk of complications and fatal outcomes after a breakthrough COVID-19 infection, and further research is needed to find more appropriate management strategies after breakthrough infections.

## Figures and Tables

**Figure 1 vaccines-10-01911-f001:**
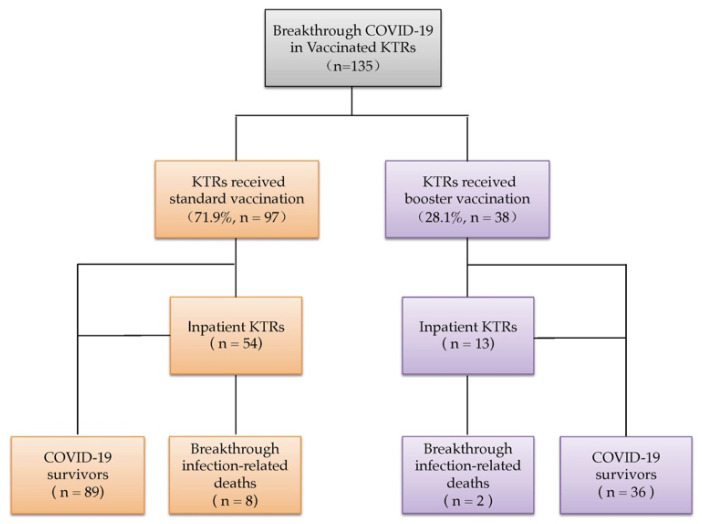
COVID-19 breakthrough infections in vaccinated kidney transplant recipients.

**Table 1 vaccines-10-01911-t001:** Center-level summary of SARS-CoV-2 breakthrough infections in KTRs.

No. of Vaccinated (*n* = 16,820)	Country	Vaccine Type						No. of Infections (*n* = 633)	No. of Dead (*n* = 75)	BI Rate (%)
		BNT162b2	mRNA-1273	Jansen	CoronaVac	ChAdOx1	Other			
1402 [18]	Canada	1048	253			48	53	3	1	0.21
341 [19]	USA	341 (mRNA vaccines > 95%)						1	0	0.30
1680 [20]	USA	1680						8	0	0.48
904 [21]	USA	658	229	17				7	1	0.8
843 [12]	Spain	843						15	1	1.78
380 [22]	USA	380						7	0	1.8
1509 [23]	Czech Rep.	1274	235					33	8	2.18
800 [24]	Spain	800						21	1	3.0
372 [25]	Greece	372						13	1	3.49
3340 [26]	Brazil				3340			135	30	4.04
1034 [27]	Spain	7	1027					44	4	4.26
164 [28]	Croatia	164						8	0	4.88
3201 [4]	Scotland	962				2239		259	26	8.09
850 [29]	Denmark	850						79	2	9.29

BI: breakthrough infection.

**Table 2 vaccines-10-01911-t002:** Case and case series of COVID-19 breakthrough infections in vaccinated KTRs.

Study	Case (*n*)	Country	Age (Y)	Sex (M/F)	Years Since KT	Maintenance IS	No. of Hospilization	Vaccine Type	No. of Dose	Days from Last Dose to COVID-19	No. of Deaths
Basic [30]	1	Croatia	46	1/0	14	TAC/MMF/Steroid	1	BNT162b2	2	14	0
Wadei [31]	1	USA	60	1/0	1.3	TAC/MMF/Steroid	1	mRNA-1273	2	44	0
Tsapepas [21]	7	USA	29–68	4/3	1.5–18	TAC/MPA/Steroid (2/7) TAC/MPA (4/7) S/MPA (1/7)	2	mRNA-1273 (2/7) BNT162b2 (5/7)	2	17–46	1
Tsalouchos [32]	1	Italy	49	0/1	NA	TAC/MMF/Steroid	NA	BNT162b2	2	23	0
Tau [5]	14	Israel	26–85	10/4	0.6–21	TAC/MMF/Steroid (11/14)	8	BNT162b2	2	25–85	4
						TAC/Steroid (2/14)					
						S/MMF/Steroid (1/14)					
Wijtvliet [33]	1	Belgium	23	0/1	1.4	S/MMF/Steroid	1	BNT162b2	2	41	0
Malinis [34]	2	USA	65–80	0/1	2.2–6.3	Bela/MMF/Steroid (1/2)	0	mRNA-1273 (1/2)	2	18–63	0
						Bela (1/2)		BNT162b2 (1/2)			
Chenxi [22]	7	USA	49–77	5/2	1.8–6.2	TAC/MMF/Steroid (6/7)	4	mRNA-1273 (2/7)	2	16–75	0
						TAC/Steroid (1/7)		BNT162b2 (5/7)			
Loconsole [35]	1	Italy	48	0/1	6.0	TAC/MMF/Steroid	1	BNT162b2	2	71	0
Aslam [19]	1	USA	67	1/0	6	CsA/Steroid	1	BNT162b2	2	72	0
Chang [36]	1	USA	73	1/0	1.3	TAC/MMF/Steroid	1	mRNA-1273	2	107	0
Mehta [20]	8	USA	36–73	5/3	0.3–20.7	TAC/MMF/Steroid (4/8) TAC/Steroid (1/8) TAC/MMF (1/8) TAC/AZA (1/8) CsA/MMF (1/8)	I3	mRNA-1273 (4/8) BNT162b2 (4/8)	2	20–77	0 ^#^
Meshram [37]	1	India	71	1/0	16	CsA/MPA/Steroid	1	ChAdOx1	2	20	1
Ali [6]	8	USA	27–72	NA	0.2–4.4	TAC/MMF/Steroid (6/8)	3	mRNA-1273 (2/8)	2	18–45	0
						TAC/MPA/Steroid (1/8)		BNT162b2 (5/8)			
						Bela/MPA/Steroid (1/8)		Jansen (1/8)			
Anjan [38]	12	USA	32–81	6/6	NA	TAC/MMF/Steroid (5/12)	5	BNT162b2 (12/12)	2	15–96	0
						TAC/MMF (5/12)					
						TAC/Bela/Steroid (1/12)					
						TAC/Bela (1/12)					
Chen [39]	1	USA	57	0/1	23	MMF/Steroid	1	mRNA-1273	2	42	0
Almaghrabi [40]	3	Saudi Arabia	42–69	2/1	5.0–27.0	TAC/MMF/Steroid (2/3)	3	BNT162b2 (2/3)	2	21–150	1
						S (1/3)		ChAdOx1 (1/3)			
Marinaki [25]	13	Greece	21–72	11/2	1.2–15.7	TAC/MMF/Steroid (11/13)	7	mRNA-1273 (3/13)	2	57–115	1
						CsA/MMF/Steroid (2/13)		BNT162b2 (10/13)			
Radcliffe [41]	4	USA	43–73	4/0	1.7–11.6	TAC/MMF/Steroid (4/4)	1	mRNA-1273 (1/4)	3	43–73	0
								BNT162b2 (3/4)			
Chung [42]	7	Singapore	34–69	3/4	1.8–31	Aza/CsA/Steroid (2/7)	7	BNT162b2	2	35–175	0
						TAC/MMF(MPA)/Steroid (2/7)					
						CsA/MMF/Steroid (1/7)					
						TAC/Steroid (1/7)					
						EVR/Steroid (1/7)					
Natori [43]	1	USA	66	0/1	10	TAC/MMF	1	BNT162b2	3	90	0
Benotmane [44]	39	France	19–79	23/16	0.1–32.9	TAC/MMF(MPA)/Steroid (23/39) MMF(MPA)/Bela/Steroid (8/39) CsA/MMF(MPA) (2/39) CsA/MMF(MPA)/Steroid (2/39) TAC/MMF(MPA) (2/39) TAC/Steroid (1/39) CsA/Steroid (1/39)	6	mRNA-1273 (22/39) BNT162b2 (15/39) mRNA-1273 & BNT162b2 (2/39)	2 (6/39) 3 (27/39) 4 (6/39)	49–351	2
Fahim [45]	1	USA	45	0/1	2.2	TAC/MMF/Steroid	1	mRNA-1273	2	120	0

M: male; F: female; IS: immunosuppression; TAC: tacrolimus; MMF: mycophenolate mofetil; MPA: mycophenolic acid; EVR: everolimus; S: sirolimus; Bela: belatacept; CsA: cyclosporin A; AZA: azathioprine; mTORi: mechanistic target of rapamycin inhibitor. ^#^ 1 died due to a cause other than COVID-19.

**Table 3 vaccines-10-01911-t003:** Comparison between KTRs who received standard vaccination and those who received booster vaccination.

Variables	All (*n* = 135)	Booster Vaccination (*n* = 38)	Standard Vaccination (*n* = 97)	*p*
Males, *n* (%) ^a^	78 (61.4)	23 (60.5)	55 (61.8)	0.89
Age (y), median [IQR]	59.0 [49.0, 69.0]	60.5 [51.0, 72.0]	58.0 [48.0, 68.0]	0.32
Years since KT (y), median [IQR] ^b^	3.9 [1.7, 8.6]	3.5 [1.5, 7.7]	4.0 [2.0, 9.4]	0.54
Maintenance IS at diagnosis of COVID-19				
Tacrolimus, *n* (%)	105 (77.8)	29 (76.3)	76 (78.4)	0.80
MMF/MPA, *n* (%)	112 (83.0)	37 (97.4)	75 (77.3)	<0.01
Steroid, *n* (%)	116 (85.9)	35 (92.1)	81 (83.5)	0.20
Belacept, *n* (%)	13 (9.6)	6 (15.8)	7 (7.2)	0.19
CNIs, *n* (%)	117 (86.7)	32 (84.2)	85 (87.6)	0.60
Triple maintenance IS, *n* (%)	98 (72.6)	34 (89.5)	64 (66.0)	<0.01
Days from last dose to COVID-19 (d), median [IQR] ^c^	74.5 [37.0, 164.8]	218.0 [149.3, 267.0]	53.0 [35.0, 91.0]	<0.01
Hospitalized, *n* (%) ^d^	67 (50.0)	13 (34.2)	54 (56.3)	0.02
Dead, *n* (%)	10 (7.5)	2 (5.3)	8 (8.3)	0.72

^a^ Data of 127 KTRs: 89 with standard vaccines and 38 with booster vaccines; ^b^ Data of 122 KTRs: 84 with standard vaccines and 38 with booster vaccines; ^c^ Data of 134 KTRs: 96 with standard vaccines and 38 with booster vaccines; ^d^ Data of 134 KTRs: 96 with standard vaccines and 38 with booster vaccines. IS: immunosuppression; TAC: tacrolimus; MMF: mycophenolate mofetil; MPA: mycophenolic acid; CNIs: calcineurin inhibitors.

**Table 4 vaccines-10-01911-t004:** Comparison between COVID-19 breakthrough infection-related deaths and survival.

Variables	BI-Related Deaths (*n* = 10)	COVID-19 Survivors (*n* = 125)	*p*
Males, *n* (%) ^a^	8 (80.0)	70 (59.8)	0.32
Age (y), median [IQR]	69.5 [64.3, 71.3]	57.5 [48.0, 68.0]	0.02
Age ≥ 60 y, *n* (%)	9 (90.0)	57 (45.6)	<0.01
Years since KT (y), median [IQR] ^b^	7.2 [2.8, 13.8]	3.9 [1.6, 7.8]	0.26
Maintenance IS at diagnosis of COVID-19			
Tacrolimus, *n* (%)	6 (60.0)	99 (79.2)	0.23
Mycophenolate, *n* (%)	9 (90.0)	103 (82.4)	1.00
Steroid, *n* (%)	10 (100.0)	106 (84.8)	0.36
Belacept, *n* (%)	2 (20.0)	11 (8.8)	0.25
CNIs, *n* (%)	7 (70.0)	110 (88.0)	0.13
Triple maintenance IS, *n* (%)	9 (90.0)	89 (71.2)	0.28
Days from last dose to COVID-19 (d), median [IQR] ^c^	69.0 [24.0, 152.5]	75.0 [39.0, 175.0]	0.49
Booster vaccination, *n* (%)	2 (20.0)	36 (28.8)	0.73

^a^ Data of 127 KTRs: 117 survivals and 10 deaths; ^b^ Data of 122 KTRs: 112 survivals and 10 deaths; ^c^ Data of 134 KTRs: 124 survivals and 10 deaths. BI: breakthrough infection; IS: immunosuppression; MMF: mycophenolate mofetil; MPA: mycophenolic acid; CNIs: calcineurin inhibitors.

**Table 5 vaccines-10-01911-t005:** Comparison between inpatients and outpatients with COVID-19 breakthrough infection.

Variables	Inpatients (*n* = 67)	Outpatients (*n* = 67)	*p*
Males, *n* (%) ^a^	39 (65.0)	34 (57.6)	0.41
Age (y), median [IQR]	65.0 [53.0, 71.0]	53.0 [43.0, 63.0]	
Age ≥ 60 y, *n* (%)	44 (65.7)	22 (32.8)	<0.01
Years since KT (y), median [IQR] ^b^	3.8 [1.6, 9.0]	4.0 [1.7, 8.2]	0.67
Booster vaccination, *n* (%)	13 (19.4)	25 (37.3)	0.02
Maintenance IS at diagnosis of COVID-19			
Tacrolimus, *n* (%)	48 (71.6)	56 (83.6)	0.10
MMF/MPA, *n* (%)	52 (77.6)	59 (88.1)	0.11
Steroid, *n* (%)	59 (88.1)	56 (83.6)	0.46
Belacept, *n* (%)	5 (7.5)	8 (11.9)	0.38
CNIs, *n* (%)	58 (86.6)	58 (86.6)	1.00
Triple maintenance IS, *n* (%)	48 (71.6)	49 (73.1)	0.85
Days from last dose to COVID-19 (d), median [IQR] ^c^	74.5 [35.0, 150.5]	77.0 [40.0, 222.0]	0.21

^a^ Data of 119 KTRs: 59 outpatients and 60 inpatients; ^b^ Data of 122 KTRs: 60 outpatients and 62 inpatients; ^c^ Data of 133 KTRs: 67 outpatients and 66 inpatients. IS: immunosuppression; MMF: mycophenolate mofetil; MPA: mycophenolic acid; CNIs: calcineurin inhibitors.

## Data Availability

Not applicable.

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
