# Peer review of "COVID-19 Breakthrough Infections in Vaccinated Kidney Transplant Recipients"

_vaccines, 2022, doi:10.3390/vaccines10111911_

Round 1

Reviewer 1 Report

Manuscript ID: vaccines-1971843

COVID-19 Breakthrough Infections in Vaccinated Kidney Transplant Recipients

In this manuscript the authors evaluate clinical protection after SARS-CoV-2 vaccination against breakthrough infections, consequential hospitalizations and deaths in Kidney Transplant Recipients (KTRs).

They analyzed n=127 Breakthrough Infections (BTIs) on patient-level with data obtained from Pub Med and Web of Science searches and differentiated between protection after standard vaccination and after administration of 3rd or 4th booster dose. More BTIs were observed after standard vaccination regime (94/127) than after receipt of 3rd or 4th booster dose (33/127). Patients who had BTI with fatal outcome (10/127, 7.9 %) were older than average. Hospitalized BTI patients (62/127, 49.2%) were older than outpatient BTIs and had received fewer booster vaccinations. Booster vaccinated KTRs with BTI were younger and the period between vaccination and infection was longer; they had lower hospitalization rates and higher rates of MMF/MPA treatment and triple immune-suppressive regimens.

Furthermore the authors analyzed data obtained from 13 kidney transplant centers via Pub Med and Web of Science searches and analyzed BTIs in 15970 fully vaccinated KTRs, showing that breakthrough infection rates varied from 0.21% to 8.09%.

These analyses contribute to existing data showing that improved protection against COVID-19 in kidney transplant recipients can be achieved by additional SARS-CoV-2 vaccine doses.  

General comments:

This work reports an analysis of data on Breakthrough Infections in Kidney Transplant Recipients on patient-level and center-level and thus contributes to the understanding of vaccine-induced protection against SARS-CoV-2 in this vulnerable group.  However, the description of the results is not clear and the discussion is too extensive.

Specific comments:

Methods:

The Methods section should state the time point when the data searches were conducted, thus to clarify until when the respective publications with patient level and center level data were included.

Results:

The results section states the occurrence and outcome of BTIs in KTRs with respect to their vaccination status, treatment schedule and demographic data and contains a table (Table 1) with the analyzed publications and the respective results according to those criteria.

To improve understanding, a flow diagram depicting the analysis of the 127 cases into respective groups with (n) and (%) should be provided. In addition, a table to summarize the characteristics of BTI with fatal outcome, hospitalized BTIs and outpatient BTIs, and similarly for standard vaccinated vs. booster vaccinated BTI would give a better overview of the results and facilitate easier understanding.

Discussion

Overall, the discussion relates to the entire topic of SARS-CoV-2 vaccination in Kidney Transplant Recipients, i. e. primary vaccine response, booster response, and protection against variants. Furthermore, safety of 3rd and 4th dose of SARS CoV2 vaccine dose, the impact of immune-suppressive treatment on vaccine responses and potential pausing of treatment prior to vaccination. These aspects relate to the findings of this work; however the discussion could be shortened.

It should also be more clearly indicated how the results obtained in this work complement or confirm the discussed information.

Potential benefits of heterologous vaccinations and pre-exposure prophylaxis are also discussed, but do not directly relate to the investigated parameters.

Line 161:

The citation for BTIs in the general population is from April 2021 (36) and is indicated as 0.01%.  It would also be clarifying to mention that this was a time point before the Omicron wave in Europe and North America, which led to much higher infection rates in vaccinated individuals.

Line 164 – 166:

When citing Bell et al (30) for data obtained in Scotland, the expression “BTI in non-vaccinated KTRs” is contradictory. If an individual is not vaccinated, no BTI can occur. Please rephrase.

Author Response

Dear reviewer,

It is so exciting to receive your letter. Thank you very much for your careful review and valuable comments on my manuscript. They are very helpful to me. We also appreciate your clear and detailed feedback and hope that the explanation has fully addressed all of your concerns. We discuss each of your comments individually along with our corresponding responses. Please see the attachment.

Reviewer 2 Report

I read the manuscript with great interest, which falls within the aim of this Journal. The topic is interesting enough to attract the readers’ attention. Regarding COVID‐19 vaccines in the transplant population, it is very important to know the efficacy in preventing the disease and at the same time the safety in terms of adverse reactions to the vaccine or risk of rejection. This study reports the clinical protection of COVID‐19 vaccines in kidney transplant recipients (KTRs), a particular population who are more prone to severe and prolonged COVID-19, highlights the need for booster doses for KTRs. The conclusions are solid and convincing, and the manuscript is of high quality.

Nevertheless, authors should clarify some points and improve the discussion, as suggested below.

 1.      What is the time range for the data? And where are the 13 centers located? it would be good to provide information on where the data were collected and from what time period. This will give the readers a sense on the scope of the study that you are reporting.

2.      These clinical data come from different articles maybe with different follow-up times, which may affect the interpretation of the clinical effect of the vaccine. It is recommended to mark follow-up time in the Table 1 and add relevant discussion. Discuss the flaws in the data.

3.      This cohort represents adult KTRs, excluding pediatric KTRs. and whether the COVID19 breakthrough infections are different from that of pediatric KTRs? Whether the authors were able to supplement covid-19 breakthrough infections in vaccinated pediatric KTRs?

4.      Will you be reporting on effects of different types of vaccines, because the efficacies are different and folks will be interested in that?

5.      What is the standard vaccine regimen? It is recommended to define it in the article.

6.      The language can be improved in many areas, for example:

Line 39 " those KTRs survived from COVID-19 " instead of " COVID-19 survivors ".
Line 50-51 " it is critical to have a broad understanding" instead of " It is very important to have a broader understanding "

Line 82 "Six-two " instead of " 62 ".

Line264 modify the word “standard”, which may be confusing.

Line 286 "strongly" instead of " mainly ".

Reviewer 3 Report

This manuscript seems a review article, not an original article. But the result is interesting to publish.

I would like to recommend changing to a "Review" article or doing further analysis of the result (pool analysis, logistic regression etc.), which would be suitable to the original article.

Major concerns.

1. Introduction section is too short. Suggest adding more information about 1) the COVID-19 vaccine failure in Kidney transplant recipients, and 2) Breakthrough infection in these recipients (rate infection, clinical manifestation, risk and life-threatening.

2. Method was not clear because many publications focusing on this scope were more than the results.

Minor concerns.

1. Line 182-184. "KTRs with a poor response might be particularly susceptible to infection by SARS-CoV-2 variants of concern (VOCs), such as the Alpha, Beta, Delta, Gamma and Omicron variants.", this sentence is redundant because VOCs were mentioned.

If you want to use the sentence I recommend rewriting such as "...(VOCs), especially in highly contagious variants such as Delta and Omicron".

Round 2

Reviewer 1 Report

Dear authors,

thank you for the revised version of your manuscript.

The respective sections of the manuskript have been changed according to the reviewer's suggestions.

Data from 6 futher studies were added after repeated database search, and a the suggested flow diagram and tables for the now 135 cases were prepared.

This revised version of the manuscript  is in my opinion suited to be published in Vaccines.

Minor corrections of English language/spelling are still required.

Reviewer 3 Report

Thank you for your substantial revision to make the manuscript suitable for publication. I am happy when saw this version, and the discussion was reasonable.

From my experience, I found some KTRs developed immunity (by seroconversion criteria) after the 5th vaccination but the level was very low. Some of them developed the immunity level same full-prime (2 doses) after the 5th dose. However, the evidence was not enough to support it.

Comments.

1. Please use only one format of p-value throughout the manuscript. This manuscript contained both p and P formats (with italicised or non-italicised).

2. Typo was found in 3. Results; "Greece (one stud)", and "Czech Rep" (this point could be using Czech Republic or Czechia.

Author Response

Dear reviewer,

It is so exciting to receive your letter again. Thank you very much for your careful review and positive comments on my manuscript.

We have studied your comments and have carefully proofread the manuscript to correct all the grammar and typos. Attached please find the revised version and relevant document, which we would like to submit for your kind consideration.

Once again, thank you very much for your comments and suggestions. And we hope that the corrections will meet with approval.

Sincerely,

Xiaojing Zhang